# In Situ Decoration of Bi_2_S_3_ Nanosheets on Zinc Oxide/Cellulose Acetate Composite Films for Photodegradation of Dyes under Visible Light Irradiation

**DOI:** 10.3390/molecules28196882

**Published:** 2023-09-29

**Authors:** Yixiao Dan, Jialiang Xu, Jian Jian, Lingxi Meng, Pei Deng, Jiaqi Yan, Zhengqiu Yuan, Yusheng Zhang, Hu Zhou

**Affiliations:** 1Hunan Engineering Research Center for Functional Film Materials, School of Chemistry and Chemical Engineering, Hunan University of Science and Technology, Xiangtan 411201, China; dyx13536505551@gmail.com (Y.D.); isxujialiang@163.com (J.X.); lingximengqr@163.com (L.M.); dengpei0901@163.com (P.D.); yuanzhengqiu@126.com (Z.Y.); zys_2002@hotmail.com (Y.Z.); hnustchemzhou@163.com (H.Z.); 2Furong College, Hunan University of Arts and Science, Changde 415000, China

**Keywords:** cellulose acetate, ZnO, Bi_2_S_3_, photocatalytic, dye degradation

## Abstract

A novel Bi_2_S_3_-zinc oxide/cellulose acetate composite film was prepared through a blending-wet phase conversion and in situ precipitate method. The results revealed that the incorporation of Bi_2_S_3_ in the film increased the cavity density and uniformity, which provided additional space for the growth of active species and improved the interaction between dye pollutants and active sites. Zinc oxide acted as a mediator to facilitate the separation of electron–hole pairs effectively preventing their recombination, thus reducing the photo-corrosion of Bi_2_S_3_. As a result, the Bi_2_S_3_-ZnO/CA composite film exhibited favorable photocatalytic activity in the degradation of various dyes. Additionally, the composite film displayed effortless separation and recovery without the need for centrifugation or filtration, while maintaining its exceptional catalytic performance even after undergoing various processes.

## 1. Introduction

The advancement of the textile industry has yielded economic affluence and societal equilibrium for humanity. Nevertheless, due to its highly polluting nature, the textile industry generates approximately 1.6 million liters of dyes per day [1,2,3]. This substantial dye consumption has resulted in a significant volume of dye wastewater, posing severe threats to both the ecological environment and human well-being [4,5]. Consequently, individuals have implemented diverse approaches, including biological, physical, and chemical methods, to effectively and safely treat dye wastewater [6]. The efficacy of conventional treatment methods is frequently constrained by the considerable persistence and solubility of synthetic dyes in water. There is a pressing need to establish economically viable and ecologically sound treatment approaches that can effectively address dye wastewater prior to its ultimate release into the environment. Photocatalytic technology, characterized by its straightforward procedural nature, gentle operating conditions, and environmentally friendly attributes, offers a viable solution by facilitating the degradation of organic pollutants in aqueous solutions into H_2_O, CO_2_, or other diminutive molecules [7,8,9]. However, the powdered photocatalyst tends to agglomerate during the process of photocatalysis, resulting in a challenge to separating and recycling. Additionally, this agglomeration can lead to secondary pollution in the water system, thereby limiting its effectiveness in degrading organic pollutants [10,11,12]. Therefore, it becomes imperative to develop an appropriate support substrate that can address these issues by facilitating the deposition of the catalyst [13].

Cellulose acetate (CA) is classified as a biopolymer, possessing notable attributes such as exceptional biocompatibility, biodegradability, robust mechanical strength, hydrophilicity, and film-forming capabilities. Consequently, it has found extensive applications in diverse sectors including medical care, packaging, textile, filtration, and various other domains [14,15,16]. However, the practical application of CA is impeded by its susceptibility to microbial attack, which poses a significant challenge. Despite the potential limitation of biofouling on CA films in water purification, the integration of nanotechnology and membrane loading technology presents a promising solution to overcome the drawbacks associated with cellulose acetate. Fu et al. [17] conducted a one-step coagulation process in Na_2_SO_4_ aqueous solutions to produce cellulose-based ZnO nanocomposite films. These films demonstrated remarkable UV-blocking properties and antibacterial activities. Similarly, Abad et al. [18] employed a phase inversion and co-precipitation method to create a CA/Au/ZnO film, which exhibited high photocatalytic activity and achieved a 95.28% degradation rate of Eosin Y pollutant. The incorporation of nanocomposite technology into fiber films can enhance their characteristics, thereby addressing their limitations and enabling their use as catalyst support substrates.

Bismuth sulfide (Bi_2_S_3_) is a promising candidate for contaminant removal, electrochemical energy conversion, and storage due to its electrical and optical properties [19,20,21]. Additionally, Bi_2_S_3_ possesses advantageous characteristics including an appropriate band gap (1.33 eV), strong visible light absorption capability, high carrier mobility, and non-toxicity [22]. Furthermore, Bi_2_S_3_ can be easily synthesized through a straightforward solution method at room temperature. However, the utilization of Bi_2_S_3_ as a sole photocatalyst encounters challenges due to its rapid recombination as a photogenerated electron–hole pair and photoinduced corrosion [23]. One approach to enhance the photocatalytic efficiency of Bi_2_S_3_ involves combining it with another photocatalytic semiconductor possessing suitable band positioning to mitigate the recombination efficiency of its photogenerated carriers. Sang et al. [24] constructed nanoflower-like Bi_2_O_3_/Bi_2_S_3_ heterojunctions that were fabricated through a one-step hydrothermal method, and obtained a removal rate of 99.72% for rhodamine B (RhB) and 91.80% for Cr(VI), respectively. Similarly, Lu et al. [25] synthesized nuclear-shell structure TiO_2_/Bi_2_S_3_ heterojunctions using the coprecipitation method, and the results indicated that an optimal quantity of Bi_2_S_3_ could enhance the photocatalytic activity of TiO_2_. The removal rate of methyl orange (MO) for TiO_2_/Bi_2_S_3_ can reach 99% within 10 min under UV irradiation. Zinc oxide (ZnO) is a typical n-type semiconductor due to its wide band gap (3.37 eV) and the low cost of raw materials that have attracted people’s attention in the field of ultraviolet detection [26,27]. The effective detection of UV light by a single ZnO is limited due to the rapid recombination of electron–hole pairs under illumination, which hinders its practical application [28,29]. It is possible to form a type-II band structure by matching the band positions between Bi_2_S_3_ and ZnO. Yuan et al. [30] successfully prepared a ZnO/Bi_2_S_3_ photocatalyst with a heterojunction structure using a solvothermal method for Cr(VI) removal and obtained a removal rate of 96% within 120 min under visible-light irradiation. It can be seen that the coupling of Bi_2_S_3_ with ZnO to form a heterostructure and stably loading it on the CA composite film present a novel and significant research idea for supported photocatalysts.

In this study, a novel stable Bi_2_S_3_-ZnO/CA composite film was successfully prepared by the blending-wet phase conversion and in situ synthesis method. SEM, XRD, XPS, and PL were used to characterize the morphology, phase structure, element valence, and electron–hole pair recombination ability of the prepared composite materials. The photocatalytic performance and stability of RhB as a model pollutant were evaluated in the presence of visible light. In addition, different dyes were tested under different conditions to determine the photocatalytic activity of the composite film, and the photocatalytic mechanism was speculated through the active species capture experiment.

## 2. Results and Discussion

### 2.1. Structural Characterization

The surface and internal structures of pure CA, ZnO/CA, Bi_2_S_3_/CA, and Bi_2_S_3_-ZnO/CA composite films were observed by FE-SEM characterization, and the results are shown in Figure 1. The surface images reveal a thickness of approximately 200 μm for the composite films. Figure 1(a_1_–a_3_) illustrate that the pure CA composite film exhibits a characteristic asymmetric structure comprising of sponge-like hollow fiber skin and irregular cavities [31,32]. Upon loading ZnO onto the CA film, as depicted in Figure 1(b_1_–b_3_), the surface of the composite film becomes smoother and the number of pores decreases significantly. Additionally, the rod-shaped ZnO is observed to extend from the interior to the exterior of the thin film, indicating a favorable integration between ZnO and CA [33]. As demonstrated in Figure 1(c_1_–c_3_), the Bi_2_S_3_ within the Bi_2_S_3_/CA composite film is observed to form a granular cluster on the film’s surface. Furthermore, the incorporation of Bi_2_S_3_ enhances the density and uniformity of the cavities within the composite film. These cavities provide additional space for the growth of active species and enhance the contact efficiency between water pollutants and photocatalysts [34]. Figure 1(d_1_–d_3_) depict the notable alterations in the morphology and distribution of Bi_2_S_3_ within the Bi_2_S_3_-ZnO/CA composite film in the presence of ZnO. The incorporation of ZnO into the composite film leads to a notable improvement in surface smoothness and a significant reduction in pore density. Consequently, the ingress of S^2−^ ions generated by thioacetamide into the pores is impeded, resulting in a lower concentration of S^2−^ ions within the ZnO/CA composite compared to the pure CA film. The elevated concentration of S^2−^ ions would promote the formation of numerous crystal nuclei and facilitate rapid crystal growth, thereby favoring the aggregation of Bi_2_S_3_ into granular clusters under high S^2−^ ion concentrations. Conversely, Bi_2_S_3_ would gradually grow at low S^2−^ ion concentrations, giving rise to the formation of flaky Bi_2_S_3_ nanosheets. Similar results have been verified in the relevant literature [35,36]. Therefore, the Bi_2_S_3_-ZnO/CA composite film displays uniform Bi_2_S_3_ micro-nano flakes (thickness of 10 nm and width of approximately 70 nm) and is evenly distributed within the fibers of the composite film. The EDS map of the Bi_2_S_3_/ZnO-CA composite film, depicted in Figure 1(e_1_–e_6_), illustrates a homogeneous distribution of Zn, O, Bi, and S elements throughout the composite film. This finding further confirms the attachment of nanostructured Bi_2_S_3_ and ZnO to the internal surface of the composite film.

The composition and crystal structure of ZnO/CA and Bi_2_S_3_-ZnO/CA composite films were determined by XRD characterization. As shown in Figure 2, the strong diffraction peaks appearing at 31.77°, 34.42°, 36.25°, 47.54°, 56.60°, 62.86°, 67.96°, and 69.01° in the ZnO/CA composite film corresponded to the (100), (002), (101), (102), (110), (103), (112), and (201) crystal planes, respectively. These characteristic peaks corresponded to the body-centered cubic structure of ZnO (JCPDS No. 36-1451) [37]. The addition of Bi_2_S_3_ resulted in a weakening of the intensity of the ZnO diffraction peaks, particularly as the Bi_2_S_3_ load increased. The Bi_2_S_3_-containing composites have quite broad diffraction peaks; this may be due to the incorporation of Bi_2_S_3_ in the film which increased the cavity density and uniformity, resulting in the ZnO and Bi_2_S_3_ being evenly distributed inside the film. Especially Bi_2_S_3_ coated the surface of ZnO, leading to a reduction in the diffraction peak intensity of ZnO. Furthermore, extremely weak diffraction peaks at 24.93° were observed, which corresponded to the (130) crystal planes of Bi_2_S_3_ [38]. The fact that the diffraction peak of Bi_2_S_3_ was very weak might be due to the high distribution and small size of Bi_2_S_3_ particles, which was consistent with findings reported in the literature [39]. These results confirmed the successful synthesis of ZnO and Bi_2_S_3_ in CA composite film using the in situ precipitate method.

Figure 3 displays the spectra of the pure CA, ZnO/CA, and Bi_2_S_3_-ZnO/CA composite films. The ZnO/CA spectrum exhibits characteristic peaks at 1747 cm^−1^, 1237 cm^−1^, 1371 cm^−1^, and 1044 cm^−1^, which correspond to N-H stretching vibration, C-H stretching vibration, -CH_2_ symmetric stretching vibration, and free C=O stretching vibration and partial H-bonded carbonyl stretching vibration, respectively [40,41]. The absorption peak at 482 cm^−1^ is attributed to ZnO, while the peak at 3442 cm^−1^ is associated with the O-H stretching mode [42,43]. The FTIR spectra of the pure CA film exhibit a marked resemblance to those of the Bi_2_S_3_-ZnO/CA and the ZnO/CA composite film, suggesting that the incorporation of bismuth sulfide and zinc oxide does not result in the formation of novel bonds with the CA film, nor does it compromise the integrity of the film’s structure.

The utilization of XPS spectroscopy facilitated the examination of the surface composition and chemical state of the Bi_2_S_3_-ZnO/CA sample, as depicted in Figure 4. The composite material is shown to contain Zn, Bi, S, O, and C elements, as demonstrated by the comprehensive spectrum presented in Figure 4a. Figure 4b displays two prominent peaks that are concentrated at 1043.65 and 1020.28 eV, with a binding energy difference of 23.37 eV, which corresponds to Zn 2p_1/2_ and Zn 2p_3/2_, respectively. These values align with the established reference values for ZnO [44]. In Figure 4c, the high-resolution X-ray photoelectron spectroscopy spectrum of oxygen is analyzed and fitted with two distinct peaks. The weaker peak observed at 530.98 eV is attributed to the solid-state lattice oxygen present in ZnO, while the stronger peak at 531.58 eV is assigned to the interaction between carbonyl oxygen atoms in CA [37]. The Bi 4f_7/2_ and Bi 4f_5/2_ orbitals corresponding to Bi^3+^ are also observed at 156.78 eV and 162.17 eV, respectively [29]. The XPS peak of sulfur observed in Figure 4e at 162.47 eV is consistent with the standard S 2p peak, providing evidence for the formation of Bi_2_S_3_. In Figure 4f, the spectra associated with C can be effectively modeled with four distinct peaks at 283.58 eV, 285.07 eV, 286.12 eV, and 287.59 eV, respectively, which correspond to H_3_C(C=O), C-H, C-OH, and C-C-O in the CA structure [45,46]. These results not only confirm the successful synthesis of Bi2S3 but also suggest no alterations have occurred in ZnO. Additionally, the absence of any other impurities on the surface of the composite film indicates the relatively high purity of the synthesized Bi_2_S_3_ nanosheets.

The optical properties of these composite films were investigated by UV-vis DRS. As illustrated in Figure 5a, both ZnO and Bi_2_S_3_ sensitization considerably broadened the spectrum of light absorption [47]. Notably, the Bi_2_S_3_ film with co-modification of ZnO and Bi_2_S_3_ exhibited superior visible light harvesting capabilities compared to the single Bi_2_S_3_ deposition, potentially due to the modulation of the bandgap by the sensitizer. The band gap width of Bi_2_S_3_-ZnO/CA composite film was calculated by the Tauc formula (ahv = A(hv − E_g_)^n^, where α, hv, A, and E_g_ were defined as the absorption coefficient, photonic energy, constant, and band gap, respectively. The n value was 1 because ZnO and Bi_2_S_3_ belong to direct bandgap semiconductors [48]. In Figure 5b, the calculated bandgap widths for ZnO/CA and Bi_2_S_3_/CA were 3.17 eV and 1.78 eV, respectively [49,50]. Additionally, the valence band (VB) and conduction band (CB) edge potentials of the ZnO/CA and Bi_2_S_3_/CA composite films were calculated using the empirical equations E_VB_ = X − E_e_ + 0.5E_g_ and E_CB_ = E_VB_ − E_g_. Here, E_VB_ and E_CB_ are the VB and CB edge potentials, respectively. E_e_ is the energy of the free electron versus hydrogen (4.5 eV), and E_g_ is the bandgap width. The X was the Mulliken electronegativity of the ZnO and Bi_2_S_3_ semiconductors, and was selected as 5.75 eV and 5.27 eV, respectively, according to the literature [49,50]. Therefore, the E_VB_ and E_CB_ values were approximately 2.84 eV and −0.34 eV (vs. NHE) for the ZnO/CA film, and 1.66 eV and −0.12 eV (vs. NHE) for the Bi_2_S_3_/CA film.

Photoluminescence spectroscopy (PL) was also utilized to examine the efficacy of electron–hole pair recombination and separation in composite films. Generally, greater fluorescence intensity denotes accelerated electron–hole pair recombination rates, whereas lower fluorescence intensity signifies heightened separation efficiency [51]. Elevated electron recombination rates are disadvantageous for photocatalytic reactions. The photoluminescence spectra of pure CA, ZnO/CA, Bi_2_S_3_/CA, and Bi_2_S_3_-ZnO/CA composite films are presented in Figure 5c. ZnO/CA and Bi_2_S_3_/CA films exhibit an absorption peak at approximately 601 nm, while Bi_2_S_3_-ZnO/CA displays a lower resolution exciton absorption peak. The results suggest that the combination of ZnO and Bi_2_S_3_ mitigates the recombination of photo-induced electron–hole pairs and enhances the efficiency of charge separation, thereby augmenting the photocatalytic activity of the catalyst.

### 2.2. Photocatalytic Efficiency

The photocatalytic performance of Bi_2_S_3_-ZnO/CA composite films was evaluated through the degradation of RhB as a model pollutant, as depicted in Figure 6a. The absence of a photocatalyst and the sole use of a pure CA film resulted in a negligible degradation of RhB, indicating that the self-degradation effect of RhB and the adsorption effect of the CA film can be disregarded. The degradation rate of single-component ZnO/CA composite film was 26.51%, which might be attributed to the good hydrophilicity of ZnO improving the adsorption capacity of RhB. The single-component 4Bi_2_S_3_/CA composite film exhibited a higher degradation rate of 50.26%, indicating that Bi_2_S_3_ has a favorable photocatalytic activity for RhB. The addition of ZnO and Bi_2_S_3_ to the CA composite film can significantly improve the photocatalytic performance, and the catalytic performance is gradually increased with the increase in the Bi_2_S_3_ loading. The highest degradation rate of 90.2% was achieved with the 4Bi_2_S_3_-ZnO/CA film as the catalyst. Notably, a further increase in Bi_2_S_3_ loading led to a decline in photocatalytic efficiency. The possible reason might be that the particle aggregation at high loading resulted in a reduction in the effective active species on the composite film and a subsequent decrease in photocatalytic performance. In the Bi_2_S_3_-ZnO/CA catalyst, Bi_2_S_3_ is deemed as the primary active constituent due to its superior photocatalytic activity compared to ZnO/CA. However, ZnO acts as a mediator to facilitate the separation of electron–hole pairs, effectively preventing their recombination, thus reducing the photo-corrosion of Bi_2_S_3_. Consequently, this mechanism significantly enhances the overall photocatalytic efficiency.

To further investigate the degradation kinetics of RhB, a pseudo-first-order kinetic model has been employed for exploration, as illustrated in Equation (3.2): ln(C_0_/C) = kt (3.2). In this equation, C_0_ and C denote the initial concentration and real-time concentration of RhB, respectively, while k represents the apparent reaction rate constant under irradiation. The results of the fitting indicated that it conformed to a quasi-first-order kinetic equation. The kinetic curves of ZnO/CA, 4Bi_2_S_3_/CA, and Bi_2_S_3_-ZnO/CA composite films are presented in Figure 6b. The apparent reaction rate constant of the 4Bi_2_S_3_-ZnO/CA composite film was determined to be 0.0175 min^−1^, which exhibited 8.8 and 4.7 times higher values compared to the single-phase ZnO/CA (0.00198min^−1^) and 4Bi_2_S_3_/CA (0.00372 min^−1^) films, respectively. These results suggested that the synergistic effect of ZnO and Bi_2_S_3_ could significantly augment the photocatalytic efficiency of the composite film. Figure 6c depicts the dynamic absorbance spectra of the RhB solution on the 4Bi_2_S_3_-ZnO/CA composite film under visible light irradiation. The absorbance at the maximum absorption wavelength (553 nm) of the RhB solution was gradually diminished over time and reached a negligible level after 120 min, indicating the degradation of RhB in the solution without any production of other derivatives during the photocatalysis process. Figure 6d displays the durability of the Bi_2_S_3_-ZnO/CA composite film in the photodegradation of RhB. The results indicated that the Bi_2_S_3_-ZnO/CA composite film exhibited excellent performance stability, as evidenced by the sustained degradation rate of RhB at 88% over five cycles. Additionally, the composite film can be retrieved from the aqueous phase without the requirement of centrifugal filtration, thereby preventing the loss of nanoparticles and minimizing the risk of secondary pollution, which is advantageous for the practical implementation of photocatalysis technology.

Figure 7a illustrates the degradation rate of RhB by a Bi_2_S_3_-ZnO/CA composite film under visible light conditions and varying pH levels. The photocatalytic removal rates for RhB were determined to be 91.98%, 90.16%, and 90.35% in acidic, neutral, and alkaline solutions, respectively, which suggested that the Bi_2_S_3_-ZnO/CA composite film exhibited commendable photocatalytic performance for RhB degradation across a broad pH range. The photodegradation of various dyes in the Bi_2_S_3_-ZnO/CA composite film was also investigated and the results are presented in Figure 7b. The removal rates of RhB, malachite green (MG), methylene blue (MB), and crystal violet (CV) were 90.16%, 88.85%, 86.88%, and 87.20%, respectively. These results suggested that the Bi_2_S_3_-ZnO/CA composite film exhibited a universal capacity for dye pollutant treatment. Furthermore, the aforementioned dyes demonstrated commendable photocatalytic efficacy in the presence of natural light (the sunlight between 2:00 p.m. and 4:00 p.m. on a sunny day with a PM 2.5 of 26 micrograms per cubic meter in Xiangtan, Hunan, China). The solar insolation was measured and recorded by an optical power meter. Notably, RhB and CV exhibited a degradation efficiency reduction of merely 2% and 4% in comparison to simulating visible light. In addition, the photocatalytic activity of the Bi_2_S_3_-ZnO/CA composite film is slightly lower than that reported in other related studies (Table 1). This may be due to the better mass transfer effect of the powder compared to the film. However, composite film materials have better practical application prospects because of their easy separation and recovery in dye wastewater treatment.

To ascertain the photocatalytic degradation mechanism, radical-trapping experiments were conducted utilizing isopropanol (IPA), triethanolamine (TEOA), and 4-hydroxy-TEMPO (TEMPO) as scavengers for hydroxyl groups (•OH), holes (h^+^), and superoxide radicals (•O_2_^−^), respectively [52,53]. Figure 8 demonstrates that the inclusion of IPA had negligible impact on the degradation of RhB, thereby suggesting that •OH played a minor role in the photodegradation of RhB by the Bi_2_S_3_-ZnO/CA composite film. Conversely, the presence of TEOA and TEMPO resulted in a substantial inhibition of the RhB degradation, indicating that h^+^ and •O_2_^−^ were the primary active components responsible for the photocatalytic degradation of RhB by the Bi_2_S_3_-ZnO/CA composite film.

Based on the aforementioned results, the potential mechanism for the photodegradation of RhB by the Bi_2_S_3_/ZnO-CA composite film was hypothesized, as illustrated in Figure 1. When exposed to visible light, both the ZnO and Bi_2_S_3_ particles present in the film were stimulated to generate electrons and holes in their respective conduction and valence bands. Due to the more negative conduction band (CB) energy of ZnO (−0.34 eV/NHE) compared to that of Bi_2_S_3_ (−0.12 eV/NHE), the photogenerated electrons in ZnO readily migrated to the CB of Bi_2_S_3_, where they were scavenged by the available surface O_2_ to produce •O_2_^−^ radicals [54]. Simultaneously, the photogenerated holes in ZnO migrated to the valence band (VB) of Bi_2_S_3_ due to the higher VB energy of ZnO (+2.84 eV/NHE) compared to Bi_2_S_3_ (+1.66 eV/NHE). The abundant concentration of reactive holes in the Bi_2_S_3_ VB effectively oxidizes the dye [55]. The strong oxidation ability of these holes and •O_2_^−^ radicals can over-decompose RhB into harmless molecules such as CO_2_ and H_2_O. As a result, the separation of electron–hole pairs is efficiently achieved, leading to a significant reduction in charge recombination and an enhancement in the photocatalytic activity of the composite film.

## 3. Experimental Section

### 3.1. Synthesis of Bi_2_S_3_-ZnO/CA Composite Films

The Bi_2_S_3_-ZnO/CA composite films were prepared by blending-wet phase conversion and in situ precipitate method [53,56]; the detailed steps are presented in Figure 2. Typically, 2 g CA, 1 g ZnO, and Bi(NO_3_)_3_•5H_2_O at different quantities (1, 2, 3, 4, and 5 g) were successively added to the 20 g DMF solvent with vigorous stirring at room temperature for 2 h. The obtained uniformly mixed solution was held in the air for 30 min to eliminate bubbles. Subsequently, the mixture was cast evenly on the glass mold, and quickly immersed in thioacetamide solutions (equimolar with Bi(NO_3_)_3_•5H_2_O) for 8 h to the in situ synthesis of Bi_2_S_3_-ZnO/CA composite films [35]. Afterward, the resultant composite films were rinsed with ethanol and deionized water several times, and subsequently dried overnight at −40 °C in a freeze-drying oven. Based on the content of Bi(NO_3_)_3_•5H_2_O, the acquired films were labeled as 1Bi_2_S_3_-ZnO/CA, 2Bi_2_S_3_-ZnO/CA, 3Bi_2_S_3_-ZnO/CA, 4Bi_2_S_3_-ZnO/CA, and 5Bi_2_S_3_-ZnO/CA, respectively. By comparison, the pure CA, ZnO/CA, and Bi_2_S_3_/CA films were also prepared using the same methods.

### 3.2. Characterization

The crystal structures of the composite films were analyzed using the X-ray diffraction (XRD) measurement on a Brucker AXS D8-Advance with Cu-K*α* irradiation. The morphology of the samples was observed by Field Emission Scanning Electron Microscopy (FE-SEM, Zeiss Sigma 300). FT-IR spectra were recorded using a Brucker TENSORII FT-IR spectrometer from 4000 to 400 cm^−1^. Surface electronic states were detected using X-ray photoelectron spectroscopy (XPS) using a K-Alpha 1063, and all binding energies were corrected by a C1s peak at 284.8 eV. The composite films’ UV-vis diffuse reflectance spectra (DRS) were obtained using a Shimadzu UV-2550 spectrophotometer. The recombination of electron–hole pairs was explored using photoluminescence spectroscopy (PL) on HitachiF-2700.

### 3.3. Evaluation of Photocatalytic Activity

The photocatalytic activities of composite films were evaluated by the degradation of dyes under visible light illumination using a 300W Xe lamp equipped with a UV cutoff filter (λ between 420 and 800 nm). Generally, 0.2 g of catalyst was added to a condensation reactor containing 200 mL of RhB (10 mg/L). The suspension was continuously stirred for 30 min in the dark to achieve an adsorption–desorption equilibrium between the composite film and RhB-simulated liquid. Then, the mixture was subjected to photocatalysis using a xenon-mercury parallel light source. During the photocatalytic reaction, 10 mL of the reaction solution was sampled every 20 min. The residual concentration of RhB was measured using a UV-visible spectrophotometer at the maximum absorption wavelength. In addition, hydroxyl (•OH), holes (h^+^), and superoxide radicals (•O_2_^−^) were detected with IPA, TEOA, and TEMPO in this photocatalytic reaction, respectively.

## 4. Conclusions

In summary, a novel Bi_2_S_3_/ZnO-CA composite film was successfully prepared through the blending-wet phase conversion and in situ precipitate techniques. The results indicated that the incorporation of ZnO induced a notable modification in the configuration and dispersion of Bi_2_S_3_ particles, transitioning from clustered nanospheres to evenly distributed nanosheets. Furthermore, ZnO acted as a mediator for the separation of electron–hole pairs, effectively impeding the recombination of photo-generated electron–hole pairs and mitigating the photo-corrosion of sulfides. Among the catalysts, the 4Bi_2_S_3_/ZnO-CA composite film showed the best photocatalytic activity with a 90.16% RhB degradation rate. More importantly, the composite film presented advantages in terms of operational simplicity, facile recovery, and the elimination of secondary pollution compared to powder-type photocatalysts, rendering it an efficient and versatile material for wastewater treatment.

## Data Availability

The data in this study are available from the corresponding author upon reasonable request.

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
