# Peer review of "In Situ Decoration of Bi2S3 Nanosheets on Zinc Oxide/Cellulose Acetate Composite Films for Photodegradation of Dyes under Visible Light Irradiation"

_molecules, 2023, doi:10.3390/molecules28196882_

Round 1

Reviewer 1 Report

Notes:

General comment: The work presents a catalyst made from bismuths sulfide and ZnO/cellulose acetate. The authors described here its synthesis and characterization and evaluation of its photocatalytic degradation efficiency for different dyes. The idea presented here is typical nanomaterial catalyst synthesis and evaluation. While I find that they presented a lot of techniques for characterization and evaluation, and that their overall research design is appropriate, I think there are a few missing control experiments that should be considered. Specifically, I have included comments on these in some of the bullet points below. Also, some of the results are not sufficient to draw out certain specific interpretation as written.

Specific comments:

·         Define # in the authorship.

·         Some typographical and grammatical errors need to be corrected.

·         Line 98-108: where are a4, b4, c4, and d4?

·         In lines 106-108: “These cavities provide 106 additional space for the growth of active species and enhance the contact efficiency be- 107 tween water pollutants and photocatalysts [34].”, it would be interesting to know if the addition of bismuth sulfide increases the surface area. Therefore, I suggest to include this experiment or information regarding this.

·         The scale bar for the inset of Fig. 1. a4d4 should be provided.

·         Why is the Bi2S3 morphology different in Bi2S3-CA and in Bi2S3-ZnO/CA?

·         The XRD of the Bi2S3-containing composites have quite broad peaks in the 20-35 degrees that it should not be confidently assigned to bismuth sulfide. Are the authors sure these are crystalline and not amorphous?

·         Fig 3. The lines are too thick, I suggest that the authors decrease the thickness for the spectra to see the details more clearly.

·         Fig. 4: did the authors also do XPS for the reference materials to confirm the difference? Sometimes, interaction could also show slight shifts in the binding energies. I suggest the authors to also include these.

·         Section 2.2.: The authors have to consider the increase in surface area with the addition of these catalysts. I suggest they do surface area determination and also consider it in evaluation of the photocatalytic efficiency.

·         Lines 201-216: The authors should mention the composition which are effective catalysts.

·         Lines 262-264: The wavelength cutoff for Xe should also be indicated in the label (to clearly show that this is for visible light). Also, what is natural light here? This should be mentioned in the body or experimental section. 

·         To support lines 337-339, the authors should show EDS for the Bi2S3-CA composite.

The language is fine. They just need to correct some typos and improve on the grammar in some parts.

Reviewer 2 Report

In this research paper report the In situ decoration of Bi2S3 nanosheets on zinc oxide/cellulose acetate composite films for photodegradation of dyes under visible light irradiation . The present form is not suitable for publication in the molecules journal. After the correction it will be suitable for publications.   

Abstract:  The abstract provides a broader overview, highlighting key findings. Its recommended to revise the abstract to ficus more on the specific findings. abstract section  

Authors should give conclude the introduction with clearly informative objectives and elaborate on these within the context of recent research. 

 Synthesis of nanosheets: Please refer to the experimental protocol used for this synthesis. There is no citation.   

Author should compared with the related work in table form.  

Author should condense the conclusion while ensuring that it encompasses the overall findings.  

Some closely related papers can be referenced to boost the discussion.  

Minor editing of English language required

Round 2

Reviewer 1 Report

·         The authors claim that they corrected a part of the text but they did not.

·         Indicate clearly that # refers to shared first-authorship.

·         Fig.1 inset: Even if these were enlargement, they should still provide a scale bar for the enlargement (it is not so difficult to do it).

·         p.1-3: typographical errors (missing spaces, etc.) and grammatical errors (lines 58, 126, 144-145) should be corrected.

·         Line 68: define e-h+ or just use “electron-hole”.

·         Abbreviations should be defined in their first usage (eg. RhB).

·         The EDS resolution is too low. I would assume that it is better (or also useful) to do it for the cross-section in which possibly the differentiation between the small and huge particles are more obvious.

·         The description of the natural sunlight should also be done in terms of airmass (AM 1 or AM 1.5) for example so that it can be repeated in other parts of the world if necessary.

·         In Table 1, one has to specify the dye/s under examination (and the pHs) since degradation kinetics can also depend on it.

·         They should mention the surface area (and their BET results) for transparency.

·         They should also include a statement on the broadness of their peak in the XRD discussion (they answered this question in my previous review but did not incorporate it in the paper).

The English is fine but they should correct typographical and grammatical errors. 
